# Sinus Augmentation for Implant Placement Utilizing a Novel Synthetic Graft Material with Delayed Immediate Socket Grafting: A 2-Year Case Study

**DOI:** 10.3390/jcm12072485

**Published:** 2023-03-24

**Authors:** Peter Fairbairn, Stuart Kilner, Dominic O’Hooley, Andrew Fish, Gregori M. Kurtzman

**Affiliations:** 1Dental Clinic, School of Dentistry, University of Detroit Mercy, 2700 Martin Luther King Jr Blvd, Detroit, MI 48208, USA; 2Private Practice, London, UK; 3Private Practice, Leeds, UK; 4Private Practice, Jersey, UK; 5Private Practice, Silver Spring, MD, USA

**Keywords:** calcium sulfate, CaSO_4_, beta tricalcium phosphate, β-TCP, synthetic graft, crestal sinus augmentation, socket preservation

## Abstract

Frequently, sinus augmentation is required when replacing failing or missing molars in the maxilla due to loss of alveolar bone related to periodontal disease, pneumatization of the sinus or a combination of the two factors. Various materials have been advocated and utilized; these fall into the categories of allograft, xenograft and synthetic materials. This article shall discuss a study of 10 cases with a 2-year follow-up utilizing a novel synthetic graft material used for sinus augmentation either simultaneously with implant placement or in preparation for sinus augmentation and implant placement in the posterior maxilla. The results of the 10 cases in the study found consistent results over the 2-year study period with maintenance of the alveolar height at the maxillary sinus. A lack of complications or failures in the study group demonstrates the technique has useful applications in increasing ridge height to permit implant placement inferior to the sinus floor.

## 1. Introduction

Frequently, sinus augmentation is required when replacing failing or missing molars in the maxilla due to loss of alveolar bone related to periodontal disease, pneumatization of the sinus or a combination of the two factors. Various materials have been advocated and utilized falling into the categories of allograft, xenograft and synthetic materials. Each category has been well documented for the utilization in sinus augmentation procedures in the literature [1,2,3]. Several factors weigh in decisions when selecting a type of osseous graft material. Allograft and xenograft materials have higher material cost than synthetic materials which may factor in the clinician’s choice of which material to use. Allografts have been reported to produce a significantly higher amount of newly formed bone than xenografts [4]. Xenograft materials have been reported to not fully resorb over time, thus leaving particles present that will be in contact with the implant’s surface and decreasing the bone to implant (BIC) percentage which may potentially affect long-term survival of the implant [5,6]. Additionally, patients may have objections to the use of allograft or xenograft materials for religious or personal reasons or fear of disease transmission from the graft material utilized [7,8]. In addition, reported risks of bovine spongiform encephalopathy (BSE) as well as complications which may include acute and chronic sinusitis, maxillary fungus ball, material displacement, immune reactions, chronic inflammation and foreign body reaction have been reported with the use of xenograft materials [6,9,10,11,12,13]. 

The reported potential issues with allograft and xenograft materials have led to the increased use of synthetic materials in dental grafting treatment including maxillary sinus augmentation. Ideally, selection of a synthetic graft material should use one that is fully resorbable over time, without residual particles remaining when the graft conversion has completed to host bone. Two such materials available are calcium sulfate (CaSO_4_) and beta tricalcium phosphate (β-TCP). Calcium sulfate has been utilized for over 120 years in graft procedures, creating a stable graft material that sets hard, acting as a cell occlusive barrier preventing soft tissue ingress into the graft during its healing and organization phase. β-TCP has also been used medically for many years as a graft material with several benefits suitable to its use in dental graft treatment. It is osteoconductive as well as osteoinductive while being fully resorbed as it is replaced by host bone [14,15,16,17]. 

CaSO_4_ is unable on its own to initiate osteogenesis, but potentially has the ability to encourage the formation of bone through its dissolution. As the CaSO_4_ undergoes dissolution, a release of resorbable Ca^2+^ ions results in a localized acidity stimulating osteoblasts while retarding osteoclastic activity. The localized acidity contributes to the antimicrobial activity [18,19]. However, on its own, it is not an effective graft material as its resorption rate is significantly faster than host bone growth. This results in an absence as an appropriate scaffold within the defect being grafted, so it has time-limited osteoconductive properties [20,21]. Evidence has also been reported, demonstrating that CaSO_4_ has some osteoinductive properties [22,23]. As the material resorbs, the induced chemical changes and physical features such as porosity influence new vessel formation [24].

A combination of calcium sulfate and beta tricalcium phosphate combines the benefits of those two materials versus when utilized individually. The hard setting of the CaSO_4_ benefit as a membrane is not required, and complements the osteoconductive and osteoinductive nature of the β-TCP while being fully resorbed over time. CaSO_4_ combined with β-TCP as a grafting material may be utilized for guided bone regeneration resulting in pronounced high-quality bone. A bioactive alloplastic bone graft, when utilized in oral and maxillofacial reconstruction, has demonstrated pronounced bone healing with a high replacement (biodegradation) rate. The potential of utilizing this combination of synthetic materials instead of bovine xenografts and allografts may decrease disease transmission risk [25]. 

β-TCP is structurally porous material which undergoes resorption over a 9- to 16-month period [26]. The resulting bone volume unfortunately may be reduced when utilized alone. The clinical use of β-TCP has been an supplemental material combined with other less resorbable graft materials or as a volume expander for autogenous bone grafts [27]. Therefore, a combination of CaSO_4_ with β-TCP enhances the individual properties of those individual graft materials. This produces a stable hardening paste that adapts to the shape of bony defect and is a porous bone substitute that serves as a scaffold for bone regeneration [28,29,30,31,32,33]. Addition of β-TCP may result in clinical improvements in terms of bone compared to other graft materials such as CaSO_4_ alone [34]. 

Clinically, a self-stabilizing graft material will reduce the need for membranes, resulting in shortened surgical treatment time, lower treatment cost and a more simplified surgical approach. In the absence of a membrane, no impediment to stromal cell-derived factor induction from the periosteum occurs as this results in the presence of mesenchymal cells at the bone healing site, which will allow those cells to differentiate into osteoblasts. Utilization of this composite graft material containing CaSO_4_ and β-TCP has demonstrated very favorable healing outcomes related to improved access of the periosteal blood supply to the graft materials without an intervening membrane that may block it [35]. The rapid resorption rate of CaSO_4_ permits neovascular ingrowth, with its improved angiogenesis. 

EthOss graft material (EthOss^®^ Regeneration, Silsden, UK) is composed of 65% βTCP and 35% CaSO_4_. Clinically, 50% new bone is observed at 12 weeks replacing the graft material and full resorption usually occurs over the following 6–12 months depending on the volume of graft present. Studies have reported these synthetic grafts not only serve as a three-dimensional scaffold for host bone growth, but also actively promote osteoinduction during bone regeneration [36]. This has proven to be a biocompatible, osteoconductive and bioresorbable bone graft substitute [29]. Histomorphometric analysis revealed that sites grafted with EthOss^®^ were occupied by 50.28% new bone, with 12.27% residual grafting material, and 37.45% connective tissue [37]. The combination of the two materials in EthOss^®^ graft material provides ease of use without the need for a membrane for guided bone regeneration procedures [38].

## 2. Methods and Materials

The 10 cases of patients with significant periodontal bone loss in the posterior maxilla and pneumatization of the maxillary sinus that were planned for treatment of implant placement were included in the study. These sites either had a mobile tooth with related bone loss or had been edentulous at the site prior to presentation and resulting loss of crestal height to allow implant placement without supplemental grafting. In cases where extraction was part of treatment, the tooth/teeth were extracted and the site was permitted to heal for 4 weeks to allow soft tissue to cover the extraction socket and ridge prior to graft placement. IRB approval was not necessary and as patients are not identified consent to publication of their individual cases was not required.

The sites were flapped to expose the crest and EthOss graft material was mixed per the manufacturer’s instructions. The EthOss syringe was filled to a specified level with sterile saline to hydrate the graft particles, then shaken vigorously to ensure complete diffusion of the saline with the graft material. A dry sterile gauze was then applied to the syringe’s nozzle and the plunger depressed to extruded excess saline and form the graft mix. The cover was removed from the syringe which was then transferred to the surgical site and extruded into the area to be grafted. A piece of dry sterile gauze was placed over the graft material and gentle pressure was exerted to compress the graft material and shape it to the defect being filled. The crestal sinus augmentation EthOss graft material was expressed into the osteotomy, followed by superior displacement accomplished with an osseodensification bur. The implant was then placed and if any crestal defect was present, an additional EthOss graft was placed followed by compression with dry sterile gauze. The flap was then repositioned and secured with sutures. Blue^®^m oral gel (Blue^®^m, Wijhe, The Netherlands) was placed on all healing sites post-surgery and the patient was given a tube of Blue^®^m to place themselves periodically over the next few days. The patients were each given post-op antibiotics, Amoxycillin 500 mg for 5 days (3 times daily) and instructed to exercise caution when chewing to avoid the site. Sutures were removed a week later and the site was inspected for healing, and the patient was told to return in 9 weeks. 

At 10 weeks post-surgery (graft placement), the site was again flapped to demonstrate new host-regenerated bone. Implant impressions were obtained and sent to the lab for fabrication of the planned screw retained restoration. 

### 2.1. Case 1

A 54-year-old female patient presented with loss of crowns on the upper left 1st and 2nd maxillary molars that had previous endodontic treatment. Examination noted minimal coronal tooth structure remaining and significant bone loss between the two molars with probing of over 10 mm. A periapical radiograph was obtained and confirmation of the poor condition of the two molars both structurally and periodontally was noted (Figure 1, left). Treatment options were discussed with the patient who wished a fixed approach. It was recommended that the two failing molars be extracted, and socket grafting be performed to create a crestal base to support an implant following site healing. At a subsequent surgery, a crestal sinus augmentation would be performed with simultaneous implant placement. This would then be allowed to heal before restoration of the implant at the 1st molar site was initiated. 

The consent form was reviewed with the patient and then signed. Local anesthetic was administered into the buccal vestibule adjacent to the molars to be treated. The two molars were atraumatically extracted, and the sockets were curetted to remove any residual tissue. The EthOss graft material syringe was held vertically and per the manufacturer’s instructions, sterile saline was added to the syringe to wet the graft material. Once the saline had wetted the graft material, the end cap was removed from the syringe and a piece of dry gauze was utilized to remove any excess saline. The syringe was then carried to the intraoral site and expressed into the sockets. The material was gently compressed to shape it to fill the void being grafted with sterile gauze for 3 min at which stage it began to harden, showing resistance to pressure and being firm to touch. The flap margins were then closed to achieve primary closure and secured with 5–0 PGA sutured in an interrupted fashion. A periapical radiograph was obtained to document the socket grafting (Figure 1, middle).

The patient returned at 10 weeks post-surgery and a periapical radiograph was obtained to evaluate the graft healing (Figure 1, right). The graft was noted to have blended with the surrounding host bone and was deemed ready for implant placement. The soft tissue covering the grafted portion of the ridge was keratinized and no inflammation was noted (Figure 2). Local anesthetic was again administered in a similar manner as the previous surgery. A crestal incision was made from the distal of the distal papilla at the 2nd premolar to a point where the 1st molar would be positioned. A releasing incision was made at the distal of the papilla and also at the posterior extent of the crestal incision and extended into the buccal vestibule. A full-thickness flap was elevated to expose the crestal ridge (Figure 3). The graft had converted to bone at the extraction sockets. A pilot drill was utilized to start the osteotomy to a depth of 4 mm, 2 mm shy of the sinus floor as measured on the radiograph. The osteotomy was increased laterally utilizing Densah osseodensification burs (Versah, Jackson, MI, USA) to a width of 4 mm (Figure 4, left). EthOss graft material that had been hydrated in the syringe was dispensed into the osteotomy and the final Densah bur was utilized to elevate the sinus floor and laterally spread the graft material to gain height for the implant placement (Figure 4, right). A 5 mm × 8 mm Paltop Addvanced (Paltop, Cesarea, Israel) implant was introduced into the site to the desired crestal depth (Figure 5, left). A cover screw was placed, and additional EthOss graft material was placed at the crest to fill a depression on the distal aspect adjacent to the implant (Figure 5, middle). A periapical radiograph was obtained to document the implant and associated sinus/crestal grafting performed at this stage (Figure 5, right).

The patient returned at 10 weeks to initiate the restoration of the integrated implant. The implant was exposed, and the cover screw was removed (Figure 6, left). An impression was obtained, and a healing abutment was placed on the site while the restoration was fabricated at the laboratory. Following return of the restoration from the lab, the patient presented. The healing abutment was removed, and a screw-retained restoration was inserted and the screw torque was adjusted to the manufacturer’s recommendation (Figure 6, middle). A periapical radiograph was obtained to verify the seating of the restoration and graft incorporation with the surrounding host bone (Figure 6, right).

A radiograph obtained at the 2-year recall demonstrated stability of the graft surrounding the implant (Figure 7). The implant and restoration remain in function with no issues reported by the patient or noted at routine recall appointments during the prior period since completion of the restoration. 

### 2.2. Case 2

A 74-year-old female patient presented with failing maxillary left 1st and 2nd molars. Examination noted grade 2 mobility on both molars. Radiographically significant periodontal bone loss was noted on the molars, and they were deemed non-salvageable (Figure 8, left). The patient was informed and the recommended treatment would require multi-staging with grafting to allow implants to be placed at both molar sites. Extraction consent forms were signed, and local anesthetic was administered. The two mobile molars were atraumatically extracted, and the sockets were curetted. The site would be allowed to heal and achieve primary soft tissue closure before the initial grafting and initiation of implant placement. The patient accepted and was appointed to return in 10 weeks.

At 10 weeks post-extraction, the patient presented, and consent forms were reviewed and signed for grafting and implant treatment. Soft tissue had healed, closing the site with keratinized tissue (Figure 8, middle). A radiograph was obtained to check what available bone was present at both sites (Figure 8, right). Sufficient height was available to place an implant at the 1st molar site in conjunction with a crestal sinus augmentation, but insufficient height was present at the 2nd molar site which would require grafting to increase crestal height and later implant placement could be performed at that site. 

Local anesthetic was administered, and a full thickness flap of the area was elevated. Utilizing a similar technique as outlined in case 1, the 1st molar site underwent a crestal lift and a 4.5 × 8.5 mm Anyridge (Megagen, Busan, Republic of Korea) implant was placed (Figure 9). The defect present at the 2nd molar site, resulted in a crestal height of 1.5 mm, which was insufficient for implant stability if a simultaneous crestal sinus left was performed (Figure 9, middle). Thus, grafting would need to be performed to prepare the site for later implant placement. EthOss graft material was hydrated in the syringe as previously described and placed into the defect to increase crestal height following healing (Figure 9, right).

The patient returned at 10-weeks implant and graft placement and the soft tissue over the posterior quadrant was noted to be healed with no inflammation present (Figure 10 left). Following local anesthetic administration a full thickness flap was elevated and the previously grafted 2nd molar crestal area was filled with immature host bone (Figure 10 right). A radiograph was obtained, which noted a crestal height of 7.1 mm present (Figure 11 left). The 2nd molar site was prepared in a similar fashion as previously detailed, utilizing Densah burs to osseodensify the area and accomplish a crestal sinus augmentation and a 5 × 8.5 mm Anyridge implant was placed (Figure 11 middle and right). An impression of the 1st molar implant was obtained to initiate its restoration while the 2nd molar implant was integrating. A healing abutment was placed on the implant at the 1st molar and a cover screw on the 2nd molar implant and a radiograph obtained to document the clinical result to date (Figure 12).

The patient returned at 10 weeks post-2nd-implant placement and a screw-retained restoration was placed on the 1st molar and a radiograph was obtained (Figure 13, left). Impressions were obtained and the restoration on the 2nd molar was placed at a subsequent appointment. Soft tissue was healthy at placement of the restoration on the 2nd molar and no marginal inflammation was noted at either site (Figure 14, left). Soft tissue at the 1-year recall remained healthy and lacking in inflammation (Figure 14, middle). At a 2-year recall, a radiograph was obtained and bone was noted to be stable at both implants (Figure 13, right). Soft tissue at the 2-year recall remained stable and no change in marginal position was noted compared to initial restoration placement (Figure 14, right).

### 2.3. Case 3

A 57-year-old male patient presented requesting replacement of a missing maxillary left 1st molar. A radiograph was obtained with a marker to evaluate the available crestal height present (Figure 15 left). The patient was informed that grafting would be needed to increase crestal height, then following healing a crestal sinus lift would be performed with simultaneous implant placement with restoration following integration.

The consent form was reviewed with the patient and signed. Local anesthetic was administered and a crestal incision was made with a flap at the site to expose the ridges top. EthOss graft material was prepared as previously described and placed into the defect on the crest to level the crest with the mesial bone distal to the premolar (Figure 15, middle).

The patient returned at 10 weeks and a radiograph was obtained to evaluate the height of the crest (Figure 15 right). A crestal sinus lift was performed following the previous steps described in case 1 and 2 and a 4 × 8.5 mm Anyridge implant was placed with a cover screw. 

The patient returned at 10 weeks and the implant was uncovered. An impression was obtained, and a healing abutment placed on the site. A radiograph was obtained to verify the healing abutment was seated, which demonstrated conversion of the graft material to immature host bone (Figure 16, right). The restoration was returned from the lab and the patient was returned to complete the treatment. The healing abutment was removed (Figure 17, left) and the restoration was inserted (Figure 17, middle). A radiograph was obtained to verify complete seating of the restoration to the implant (Figure 17, right). Radiographically, the previously grafted area at the crest and sinus blended well with the surrounding host bone.

## 3. Results

As an example, 3 of the 10 cases are presented in detail to illustrate the technique and the results of the 10 cases in the study. No failures were noted in the 10 cases in the study over the 2-year period. For case 5 and 9, periodontal probing of 3–4 mm was noted on the distal of the implant at follow-up after restoration of the implant. Both instances were treated by reinforced homecare improvement and continued on routine recall prophy scheduling (Table 1).

## 4. Discussion

As in all cases with severe periodontal tooth loss in the posterior maxilla, both the patient and their dentist often attempt to retain the teeth for as long as possible, often utilizing scaling and root planning in an attempt to mitigate the periodontitis present and preserve the tooth. This frequently leads to more severe bone loss around the affected tooth complicating the treatment needed to replace that failing tooth with an implant. Following extraction, typically the site is left to heal for a 4-month period whether socket preservation is performed at the time extraction or not prior to implant placement. When socket preservation is not performed at the time of extraction due to an inability to achieve soft tissue closure of the socket, a delay allows soft tissue to cover the site before socket grafting is performed. Socket preservation limits that ridge resorption as the area heals following extraction. Additionally, sinus pneumatization related to site healing is limited so that crestal height is not further lost. So, regeneration procedures of the extraction socket and ridge are critical for improved long-term success of the case and this allows for reduced sinus augmentation height required for implant placement.

A critical question is about what happens histologically to the graft and at what stage during healing. For example, when delayed socket grafting was performed at a site where a large defect was present following extraction (Figure 18, left), a healing period of 3 weeks post-extraction was needed, allowing primary closure post-graft. This gives us more predictability and simpler surgery, plus two attempts to clean the site. EthOss graft material was placed into the socket (Figure 18, middle) and primary closure with the flap was achieved and secured with sutures (Figure 18, right). Intraoral physical exam and radiographic measurement showed all sites to have regenerated vertically 3–6 mm and horizontally 4–6 mm with new host bone allowing for successful placement of the implants. A core sample was obtained from one case to confirm the series to be in line with earlier findings. Histology was performed by core biopsy at 10 weeks following graft placement and demonstrated well-preserved reactive (woven) trabecular bone with intertrabecular tissue composed of uniformly collagen-rich myofibroblastic tissue and 60% of the core consisting of bone (Figure 19).

Analysis of the 10 cases utilizing EthOss graft material in relation to sinus augmentation in conjunction with implant placement to increase crestal bone height that were then tracked for a 2-year follow-up found predictable results (Table 1). The cases were all loaded at 10 weeks following implant placement and have all been in function for 2 years without any failures. They were evaluated both radiographically and clinically to assess improvement in hard tissue during functional re-modelling along with the resultant improved soft tissue. Continued hard tissue improvement was noted radiographically over the 2-year study period indicating that the EthOss graft material converts to host bone, and is fully resorbed during that replacement period. Stability of the underlying grafted bone that the implant is in contact with or at the crestal area provides a stable platform for the overlying soft tissue and long-term maintenance of the associated periodontal health. 

The protocol utilized with the EthOss graft material as outlined in the case examples found that the resorption (replacement) rate of the material at 10–12 weeks was 50% new host bone (immature) and 10% residual graft particles as evidenced by histological analysis [37]. That remaining 10% of graft particles will fully resorb over the following 10 months, dependent on host physiology as reported in extensive research into porous β-TCP. As graft material resorbs, new host bone can fully turn over in function when implants are placed and loaded [39]. The key to the EthOss protocol as described is up-regulated host healing as reported in a recent study [25]. This study, utilizing Osteoprotegrin staining which is used to assess up-regulation, found that high quality bone was formed permitting implant placement sooner or the quicker loading of the implants placed with a crestal sinus augmentation technique utilizing EthOss graft material. 

## 5. Conclusions

The benefits of regeneration of the ridge with a delayed (4-week) socket graft in severely periodontally affected cases using these next generation synthetic graft materials has demonstrated both vertical and horizontal hard tissue dimensional improvement. This two-stage protocol enables a more predictable outcome and results in a very acceptable result aesthetically and functionally.

This protocol along with the newer tools for internal sinus augmentation as well as the new synthetic particulate graft material, EthOss, appears to be helpful in the restoration of these difficult cases. The protocol resulted in significant hard tissue regeneration and improved implant stability permitting implant loading at 12 weeks post-implant insertion [40,41]. The results found that the angiogenic induction of CaSO_4_ combined with β-TCP showed the highest potential, without potential issues that have been reported with allograft and xenograft materials. 

## Figures and Tables

**Figure 1 jcm-12-02485-f001:**
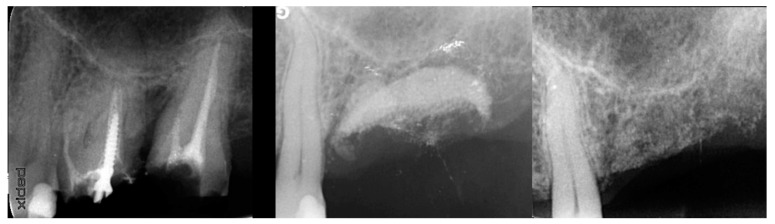
Failing endodontically treated 1st and 2nd molars (**left**), socket grafting with EthOss at time of extraction (**middle**) and following 10 weeks of healing demonstrating increased ridge height for implant placement with simultaneous crestal sinus augmentation (**right**).

**Figure 2 jcm-12-02485-f002:**
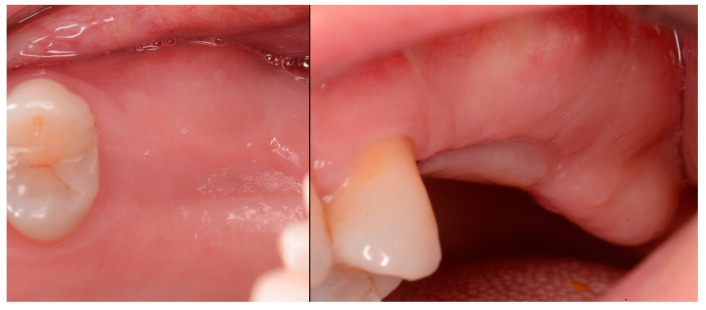
Soft tissue healing with keratinized gingiva covering the ridge at the socket grafted sites at 10 weeks healing.

**Figure 3 jcm-12-02485-f003:**
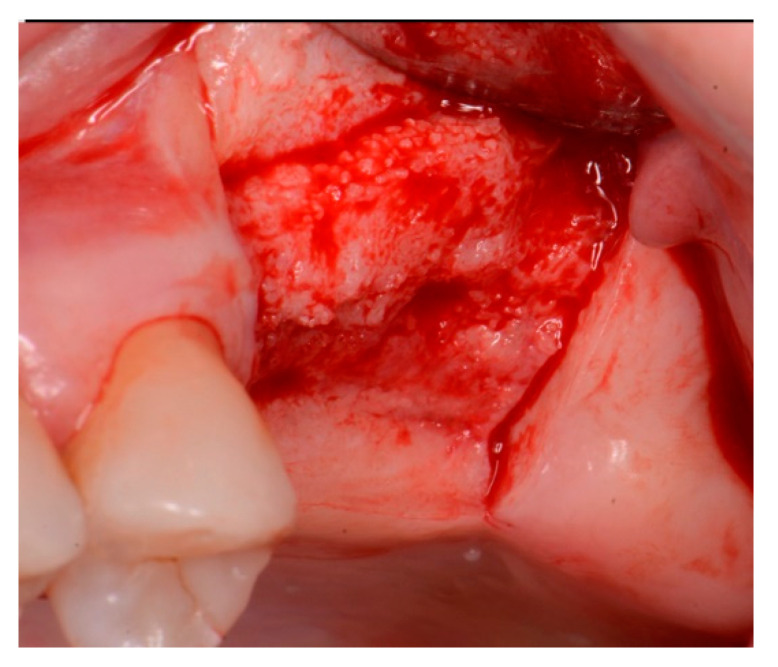
Grafted site following flap elevation and initial osteotomy demonstrating osseous fill of the grafted extraction sites.

**Figure 4 jcm-12-02485-f004:**
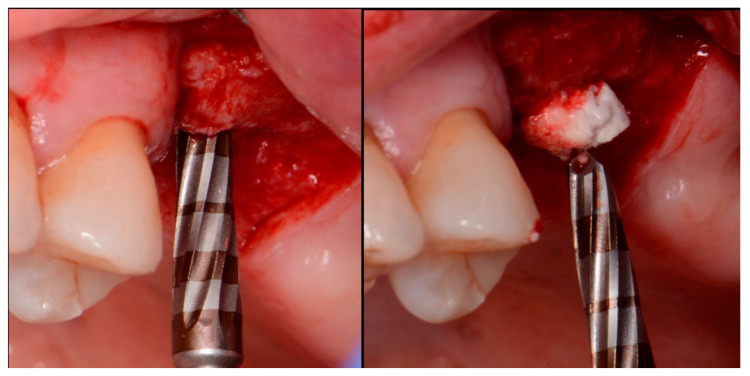
Osteotomy preparation short of the sinus floor in preparation for a crestal sinus augmentation (**left**) and utilization of the osteotomy drill to force EthOss graft material to elevation of the sinus area (**right**).

**Figure 5 jcm-12-02485-f005:**
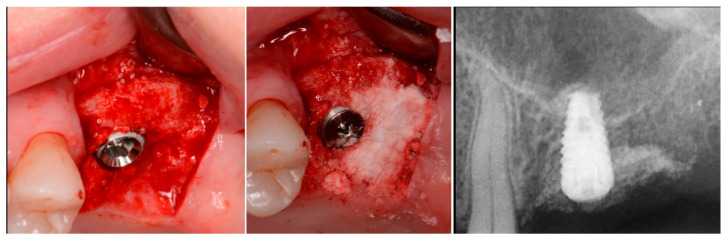
Implant placement into the crestally elevated sinus site (**left**), additional EthOss graft material to fill a crestal concavity (**middle**) and a radiograph following implant placement and flap closure (**right**).

**Figure 6 jcm-12-02485-f006:**
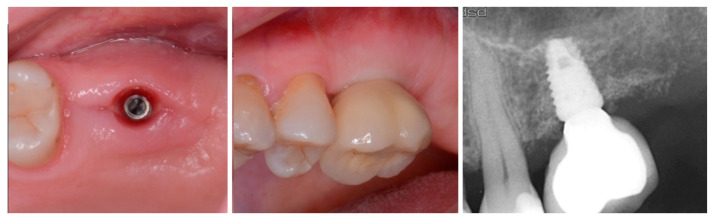
Exposure of the integrated implant after 10 weeks of healing (**left**). Screw-retained restoration placement (**middle**) and a radiograph to document seating of the restoration at the implant connector demonstrating conversion of the graft material to host bone (**right**).

**Figure 7 jcm-12-02485-f007:**
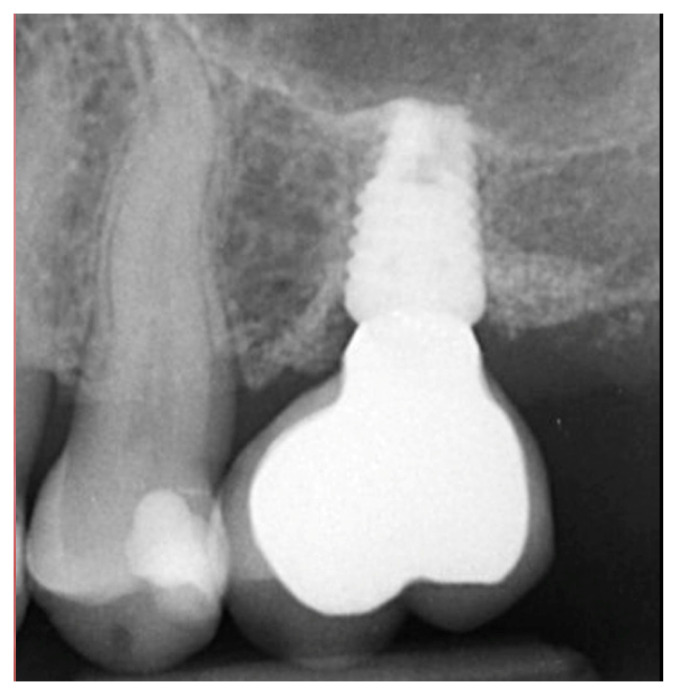
Radiograph at 2 years post-restoration placement demonstrating stability and maintenance of the grafted area.

**Figure 8 jcm-12-02485-f008:**
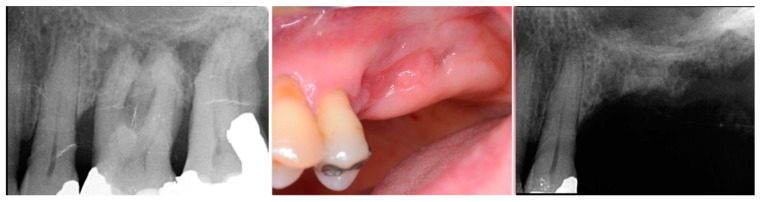
Failing 1st and 2nd molars related to periodontal bone loss (**left**), the site following 10 weeks of healing to allow soft tissue closure over the ridge (**middle**) and a radiograph demonstrating the available bone between the crest and sinus for implant placement (**right**).

**Figure 9 jcm-12-02485-f009:**
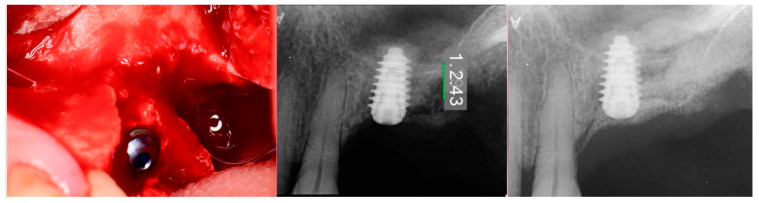
Site was reentered at 10 weeks post-extraction and an implant was placed with a crestal sinus augmentation utilizing EthOss graft material at the 1st molar site and a large defect was present at the 2nd molar site (**left**), with a radiograph obtained to document the implant placement at the 1st molar and grafting of the defect (**middle**) and following crestal grafting of the 2nd molar site (**right**). Green line is the measurement of the height of the bone at that point between the crest and the sinus floor.

**Figure 10 jcm-12-02485-f010:**
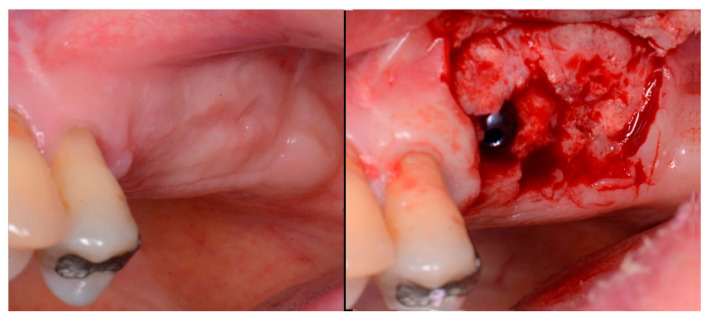
The keratinized gingiva at 10 weeks post-surgery (**left**) and following flap of the site demonstrating conversion of the osseous graft at the 2nd molar site that will allow implant placement at that site (**right**).

**Figure 11 jcm-12-02485-f011:**
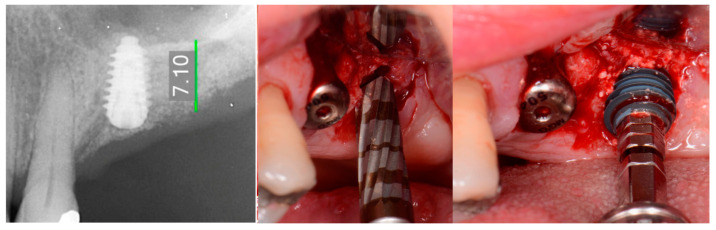
Radiograph demonstrating sufficient height of the ridge to allow primary stability of an implant placed at the 2nd molar with a crestal sinus augmentation at 10 weeks of graft healing (**left**), site preparation for the crestal sinus augmentation (**middle**) and implant placement into the site (**right**). Green line is the measurement of the height of the bone at that point between the crest and the sinus floor.

**Figure 12 jcm-12-02485-f012:**
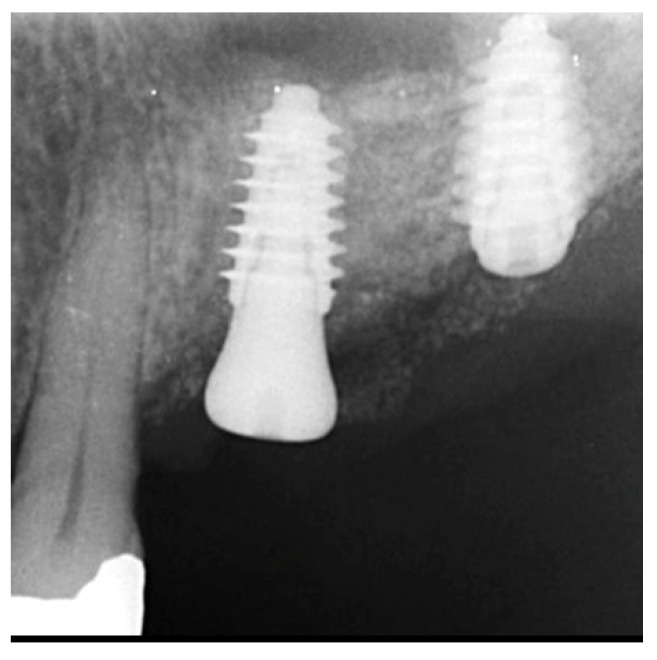
Radiograph following implant placement into the 2nd molar site with simultaneous crestal sinus augmentation.

**Figure 13 jcm-12-02485-f013:**
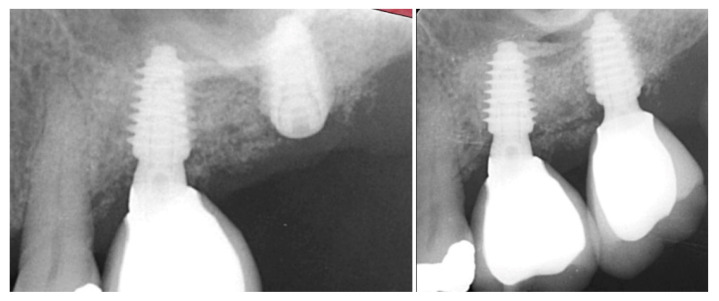
Radiograph at 10 weeks implant integration at the 2nd molar (**left**) and two years following restoration of the implant at the 2nd molar showing the improved vertical regeneration between the implants (**right**).

**Figure 14 jcm-12-02485-f014:**
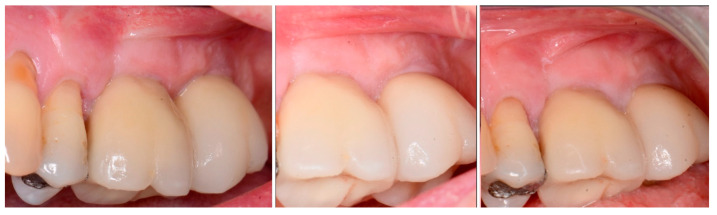
Soft tissue at placement of the restoration on the 2nd molar (**left**), at 1 year post-restoration (**middle**) and at 2 years post-restoration (**right**) demonstrating maintenance of the keratinized tissue long term.

**Figure 15 jcm-12-02485-f015:**
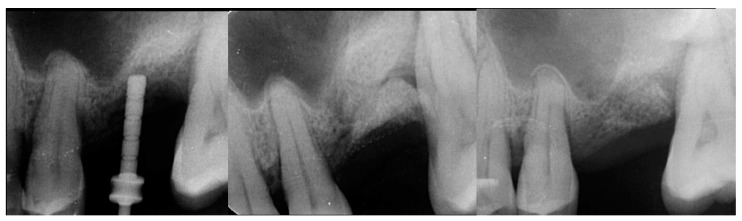
Radiograph of the initial site in preparation for implant treatment demonstrating minimal bone height available with a defect at the distal aspect of the site (**left**), of the site following EthOss grafting of the crestal graft (**middle**) and following 10 weeks of graft healing (**right**).

**Figure 16 jcm-12-02485-f016:**
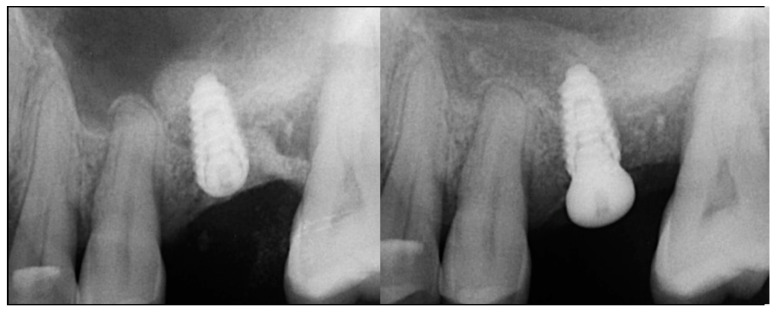
Radiograph following crestal sinus augmentation with EthOss and implant placement (**left**) and at uncovery and healing abutment placement after 10 weeks of implant placement demonstrating conversion of the graft material to host bone (**right**).

**Figure 17 jcm-12-02485-f017:**
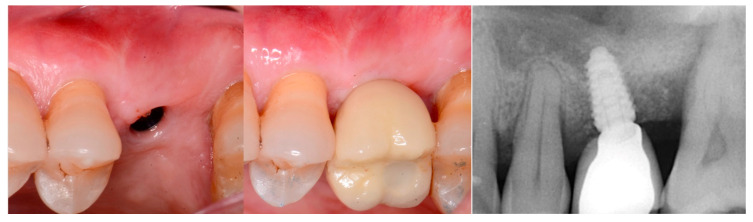
Healing abutment removal demonstrating noninflamed keratinized tissue over the grafted area (**left**), placement of the screw retained restoration (**middle**) and a radiograph to verify mating of the restoration to the implant at the connector (**right**).

**Figure 18 jcm-12-02485-f018:**
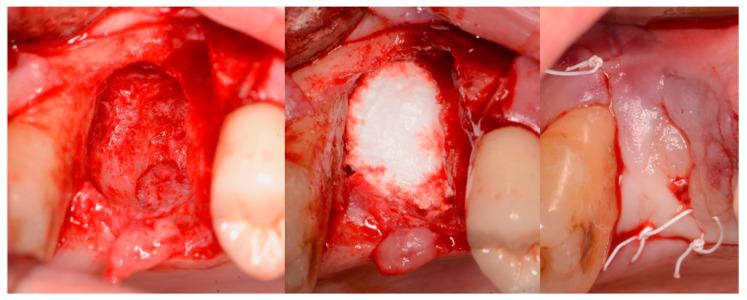
Delayed socket grafting with flap exposure of the site (**left**), placement of EthOss graft to fill the defect (**middle**) and flap placement to achieve primary closure (**right**).

**Figure 19 jcm-12-02485-f019:**
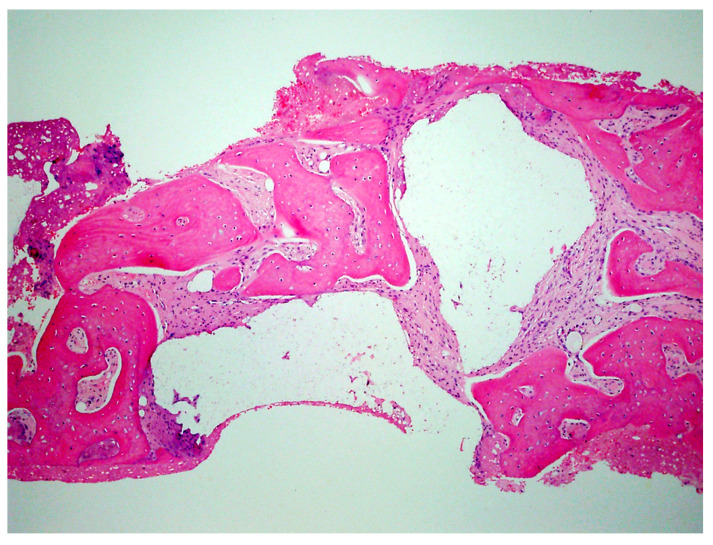
Histology of the core sample obtained following 10 weeks of graft site healing demonstrating well preserved reactive (woven) trabecular bone with intertrabecular tissue composed of uniformly collagen-rich myofibroblastic tissue and 60% of the core consisting of bone.

**Table 1 jcm-12-02485-t001:** Case studies utilizing EthOss graft of 10 patients with 2-year follow-up.

Case No	Gender	Age (Years)	Smoker (Y/N)	Perio Tooth Loss	Flap Raised (Y/N)	Quantity in Socket Graft	Horizontal Measurement at Placement	Vertical Measurement at Placement	Type of Implant Placed	NCM at Placement	Sinus Augmentation Protocol	Loaded at 12 Weeks	Loaded Vertical Measurement	2 Year Post op Vertical, Measurement of PA X-ray	2 Year post op PES Score and Papillae	Complications
1	F	54	N	Yes	Y	1 cc	8.6 mm	6.6 mm	Paltop	70 Ncm	Internal D/V	Y	9.3 mm	9.2 mm	13	Nil
2	F	64	n	Yes	Y	1 cc	7 mm	5 mm	Anyridge	65 Ncm	Internal D/V	N14 weeks	8.8 mm	9.1 mm	12	Nil
3	M	80	N	Yes	Y	1.5 cc	7 mm	9.5 mm	Paltop	50 Ncm	Internal D	Y	12mm	12.3 mm	12	NIl
4	M	82	P	Yes	Y	1 cc	6.5 mm	8.6 mm	Dio SM	50 ncm	Internal D	Y	11.1mm	10.9 mm	11	NII
5	M	79	N	Yes	Y	1 cc	9 mm	8.5 mm	Anyridge	40 ncm	Internal D/V	Y	12.4mm	11.9 mm	12	Perio, distal
6	F	43	N	Yes	Y	1 cc	10 mm	7 mm	Dio SM	45 ncm	Internal D	Y	8.2mm	9 mm	12	Nil
7	F	75	P	Yes	Y	1 cc	7.5 mm	7.1 mm	Anyridge	40 ncm	Internal V	Y	9.6mm	10.5 mm	14	Nil
8	F	81	N	Yes	Y	1 cc	8 mm	8 mm	Anyridge	55 ncm	Internal D	Y	10.7mm	11.2 mm	9	Nil
9	M	57	N	Yes	Y	0.5 cc	7 mm	6.4 mm	Anyridge	45 `ncm	Internal D/V	Y	11.65mm	11.4 mm	10	Perio, distal
10	F	92	Y	Yes	Y	1 cc	6 mm	7 mm	Anyridge	45 `ncm	Internal D	Y	9mm	9.2 mm	9	Nil

Cc= cubic centimeters, mm= millimeters, Ncm = newton centimeters, D= direct, V= vertical, Nil= none.

## Data Availability

Available from the authors upon request.

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
