# Peer review of "Sinus Augmentation for Implant Placement Utilizing a Novel Synthetic Graft Material with Delayed Immediate Socket Grafting: A 2-Year Case Study"

_jcm, 2023, doi:10.3390/jcm12072485_

Round 1
Reviewer 1 Report
The publication is well written and easy to follow. The 3 presented cases are well and in detail described to easily follow treatment and follow-up procedures and the situation at each visit.
Materials and Methods: please add information about IRB approval (given/not necessary) and/or information about patients' consent regarding publication of these data.
I would recommend not to repeat material preparation when there is already an exactly similar description in the material/methods section (case 1). Just refer to the corresponding chapter and focus on patient specific interventions and/or issues.
Please add a chapter title (Results) above the described cases. I recommend to add one sentence that only 3 cases from the 10 are to be presented in detail as representative cases. Otherwise the reader will be wondering why the remaining 7 cases are not presented in a similar fashion.
Add a result chapter with a summary of all 10 cases, describing the outcomes and, if necessary, the issue arisen using this method. You could refer in this section to Table 1.
Table 1: I urgently recommend to changes "Case no" to sequential numbering instead of initials. These are currently in the best of cases pseudonymisized and not anonymized as they should be!
Figure 19, please add scale bar and, if possible, information about the orientation in vivo (where is residual bone to be expected, etc., where is the crestal part of the biopsy). This way it is not as easy to judge the regeneration capacity of the procedure.
Please take care for consistency with some words / abbreviations , e.g. ß-TCP or ßTCP; or in Table 1: Ncm, ncn, `ncm, or distal/Disatl
Please check text style in line 156 and 338. It seems these are more comments than planned to be part of the text, at least at line 338...?
Author Response
Spelling reflects American spelling as the author who wrote the draft is American, although the other authors are UK based.
Added "IRB approval was not necessary and as patients are not identified consent to publication of their individual cases was not required."
On line 242 it already states "Utilizing a similar technique as outlined in case 1 the 1st molar site underwent a crestal...." and on line 309 under Case 3 it states "A crestal sinus lift was performed following the previous steps described in case 1 and 2 and..."
Added this above the case descriptions "Results: As an example, 3 of the 10 cases are presented in detail to illustrate the technique and the results of the 10 cases in the study."
Table 1: I urgently recommend to changes "Case no" to sequential numbering instead of initials. These are currently in the best of cases pseudonymisized and not anonymized as they should be! COMPLETED
Figure 19, please add scale bar and, if possible, information about the orientation in vivo (where is residual bone to be expected, etc., where is the crestal part of the biopsy). This way it is not as easy to judge the regeneration capacity of the procedure. Unfortunately scale bar not available and the image is to just show that graft material has converted to immature host bone.
Please take care for consistency with some words / abbreviations , e.g. ß-TCP or ßTCP; or in Table 1: Ncm, ncn, `ncm, or distal/Disatl Corrected
Please check text style in line 156 and 338. It seems these are more comments than planned to be part of the text, at least at line 338...? Corrected
Reviewer 2 Report
Thank you very much for the opportunity to review this very interesting manuscript with an important clinical application.
I have a few commentaries that I believe can help improving the manuscript.
1.General:
- Add 2 or 3 statement in the end of abstract summarizing main findings
- Add the magnification of histologic image and move it to Materials and Methods or include a chapter “Case Reports”
2.Materials and Methods (better change to case reports)
- Any bleeding, probing depth in reported cases?
3.Discussion must be improved:
- Move figures and table to Materials and Methods or include a chapter “Case Reports”
- I sense a lack of references here/discussion with previous studies
- Include indications and contraindications of the technique/material and references
- Include vantages and disadvantages related to other techniques/materials and references
4.Any negative results by using this technique/material? Any clinical case that failed and the reason of failure to be included in this report? This will enrich the content and help to improve discussion.
Author Response
English in the draft was American English as the author who wrote the draft is American, although the other authors are from the UK
Add the magnification of histologic image scale bar not available
Add 2 or 3 statement in the end of abstract summarizing main findings Added "The results of the 10 cases in the study found consistent results over the 2-year study period with maintenance of alveolar height at the maxillary sinus. A lack of complications demonstrates the technique has useful applications in increasing ridge height to permit implant placement inferior to the sinus floor."
Materials and Methods (better change to case reports) We feel keeping it as Materials and methods is more appropriate but defer to the editor on this.
Any bleeding, probing depth in reported cases? The focus of the article is the grafting and implant placement in relation to the sinus. Authors do not feel that routine probing of implants should be done unless gingival inflammation is noted or radiographic evidence of bone loss. Of the 10 cases none demonstrated gingival issues or bleeding so no probing was performed during the 2 year study period.
Discussion must be improved:
- Move figures and table to Materials and Methods or include a chapter “Case Reports” Authors feel that is Editors decision and we have no issue should they chose to do that
- I sense a lack of references here/discussion with previous studies 42 lit reference were in the article and several of those (8) are from the primary author on this same graft material
- Include indications and contraindications of the technique/material and references The authors are unaware of any specific contraindications and the article discussed the indications to this being inadequate alveolar height in relation to the maxillary sinus and its use conjuction with a crestal sinus augmentation approach. The 42 lit references support what the authors are saying.
Include vantages and disadvantages related to other techniques/materials and references We discussed use of xenografts and allografts for this and included lit references for that in the intro
Any negative results by using this technique/material? Any clinical case that failed and the reason of failure to be included in this report? This will enrich the content and help to improve discussion. We added "A lack of complications or failures in the study group demonstrates the technique has useful applications in increasing ridge height to permit implant placement inferior to the sinus floor."
Round 2
Reviewer 1 Report
The authors has take care of most of the issues stated in the first review of the manuscript. However, the most important issue is still missing a result chapter with a summary of the outcome of all (!) 10 cases. See also comment of first review:
"Add a result chapter with a summary of all 10 cases, describing the outcomes and, if necessary, the issue arisen using this method. You could refer in this section to Table 1."
This should be added before the manuscript is ready. Otherwise the manuscript should only refer to the 3 cases presented.
Author Response
In response to the reviewers comment the following was added
?Results:
As an example, 3 of the 10 cases are presented in detail to illustrate the technique and the results of the 10 cases in the study. No failures were noted in the 10 cases in the study over the 2-year period. Case #5 and 9 periodontal probing of 3-4mm was noted on the distal of the implant at follow-up after restoration of the implant. Both instances were treated by reinforced homecare improvement and continued on routine recall prophy scheduling. (Table 1)"
